# Subcellular Localization Guides eNOS Function

**DOI:** 10.3390/ijms252413402

**Published:** 2024-12-13

**Authors:** Leticia Villadangos, Juan M. Serrador

**Affiliations:** Interactions with the Environment Program, Immune System Development and Function Unit, Centro de Biología Molecular Severo Ochoa (CBM), Consejo Superior de Investigaciones Científicas (CSIC)—Universidad Autónoma de Madrid, 28049 Madrid, Spain; leticia.villadangos@estudiante.uam.es

**Keywords:** eNOS, nitric oxide, subcellular targeting, Golgi

## Abstract

Nitric oxide synthases (NOS) are enzymes responsible for the cellular production of nitric oxide (NO), a highly reactive signaling molecule involved in important physiological and pathological processes. Given its remarkable capacity to diffuse across membranes, NO cannot be stored inside cells and thus requires multiple controlling mechanisms to regulate its biological functions. In particular, the regulation of endothelial nitric oxide synthase (eNOS) activity has been shown to be crucial in vascular homeostasis, primarily affecting cardiovascular disease and other pathophysiological processes of importance for human health. Among other factors, the subcellular localization of eNOS plays an important role in regulating its enzymatic activity and the bioavailability of NO. The aim of this review is to summarize pioneering studies and more recent publications, unveiling some of the factors that influence the subcellular compartmentalization of eNOS and discussing their functional implications in health and disease.

## 1. Introduction

Nitric oxide (NO) is a non-canonical signaling molecule with multiple molecular targets. It is produced in cells by NO synthases (NOS), which generate NO through a redox reaction that requires L-arginine and molecular oxygen (O_2_) as substrates and reduced nicotinamide-adenine-dinucleotide phosphate (NADPH) and (6R-)5,6,7,8-tetrahydrobiopterin (BH4) as cofactors [1]. Seminal studies have shown that NO is involved in both physiological and pathological processes, acting as a signaling molecule in vasodilation, neurotransmission, and immune responses [2,3,4].

There are three known isoforms of NOS: neuronal (n)NOS/NOS1, inducible (i)NOS/NOS2, and endothelial (e)NOS/NOS3 [5]. Although eNOS is mainly expressed in endothelial cells, it is also present at lower levels in cardiomyocytes, platelets, erythrocytes, and renal and immune cells [6,7,8,9,10,11]. Some of the well-known roles of eNOS in organisms include the maintenance of vascular homeostasis, control of angiogenesis, platelet aggregation, and cell adhesion [12]. Moreover, its dysregulation has been implicated in the pathogenesis of several cardiovascular diseases, such as hypertension, atherosclerosis, and diabetes [13,14,15], highlighting its importance in key pathophysiological processes.

Given the highly reactive, short-lived nature of NO and its pleiotropic actions in the organism, it seems essential that NOS functions remain highly regulated. NOS binding to calmodulin (CaM) is required for maximum enzyme activity, but only nNOS and eNOS strictly depend on calcium (Ca^2+^) concentration for their activation [16]. While much attention has been given to the regulation of eNOS by Ca^2+^, phosphorylation, and other posttranslational modifications (PTMs), increasing evidence has highlighted the importance of intracellular localization in controlling its enzymatic activity and the bioavailability of NO [17]. Importantly, multiple studies have shown that the localization of eNOS within specific subcellular compartments plays a crucial role in its function [18,19].

eNOS requires the participation of various regulatory factors to ensure its trafficking across organelles and membrane domains. These dynamics are mainly regulated by a combination of diverse PTMs and protein–protein interactions. However, there are still many unknowns regarding how the complex regulation of eNOS interorganelle trafficking takes place and which role plays in the function of NO

In this review, we present pioneering studies and the most recent findings on the mechanisms underlying the intracellular translocation of eNOS, outlining our current understanding of this process and discussing its effects on NO-mediated cellular signaling.

## 2. Mechanisms Driving the Subcellular Targeting of eNOS

The role of NO as a signaling molecule in the cardiovascular system has been extensively studied [2,20,21,22]. The discovery of NOS as enzymatic cellular sources of NO and the identification of eNOS as the specific enzyme involved in NO-mediated vasodilatation propitiated further research on this protein, focusing on the regulation of its enzymatic activity and its role inside endothelial cells [23,24,25,26]. Promptly, it became clear that the localization of eNOS played a key role in its intracellular function, as this molecule was able to localize within distinct cellular organelles [27,28]. Furthermore, mutation of specific residues of eNOS disrupted its intracellular location and enzymatic activity [19,29].

The machinery that regulates eNOS intracellular trafficking includes a wide range of signaling pathways, protein–protein interactions, and PTMs [30]. Additionally, eNOS expression can be modulated by a broad diversity of stimuli, adding a whole new layer of regulation to the synthesis of NO [31]. In this section, we summarize the main mechanisms behind the intracellular localization of eNOS, focusing on the most recent findings on this topic.

### 2.1. Modulation of eNOS Expression: Transcriptional and Posttranscriptional Mechanisms

In contrast to iNOS, whose expression is mainly induced by pro-inflammatory cytokines and bacterial endotoxins, eNOS was initially thought to be a constitutive NOS isoform whose activity is regulated by PTMs. However, more recent studies have shown that the expression levels of eNOS can also be transcriptionally modulated in response to a plethora of stimuli, including fluid shear stress, cyclic stretch, reactive O_2_ species (ROS), exercise, estrogens, statins, lipids such as sphingosine 1 phosphate (S1P), and growth factors (e.g., vascular endothelial growth factor [VEGF] and transforming growth factor-β [TGF-β]) [32,33,34,35,36,37,38,39]. On the other hand, pro-inflammatory stimuli such as oxidized LDL (oxLDL), tumor necrosis factor (TNF)-α, lipopolysaccharide (LPS), or hypoxia can reduce eNOS mRNA levels [40,41,42,43].

More recently, mechanisms regulating the stability of eNOS mRNA have been identified, such as certain RNA-binding proteins or microRNAs (miRNAs) and long noncoding RNAs (lncRNAs), whose cellular expression patterns are increasingly being used as clinical biomarkers of disease [44]. The characterization of miRNAs and lncRNAs as small and long noncoding RNAs, respectively, unveiled a new world of posttranscriptional-regulating mechanisms that affect the expression of numerous genes in diverse ways. Whereas regulation by miRNAs mainly leads to translational repression, lncRNAs display a variety of modulatory mechanisms, although many of them are yet to be characterized [45]. It has been reported that under laminar shear stress conditions, eNOS mRNA stability increases, whereas oxidative stress leads to destabilization [46]. Specific miRNAs, such as miR-155, miR-222/221, miR-92a, or miR-200b, have been shown to negatively regulate eNOS by binding to its 3′ untranslated region, thereby promoting mRNA degradation and translational repression [46,47,48,49]. Moreover, new studies have identified spliced-transcript endothelial-enriched lncRNA (STEEL) and enhancer-associated lncRNA (LEENE) as additional regulators of eNOS expression in endothelial cells, influencing important transcription factors such as Krüppel-like Factor 2 (KLF2) or forming associations with chromatin at the eNOS locus during angiogenesis and as a response to shear stress [50,51,52].

### 2.2. Posttranslational Modifications of eNOS

The identification of eNOS as a membrane-associated protein promoted the study of the mechanisms underlying its subcellular targeting, leading to the exploration of the PTMs controlling eNOS localization and activity. The most studied PTMs of eNOS include acylation and phosphorylation, which are responsible for their translocation to the plasma membrane (PM) and activity, respectively [53]. Nonetheless, other eNOS PTMs have been described, including S-nitrosylation, glutathionylation, acetylation, or glycosylation [54].

#### 2.2.1. Palmitoylation and Myristoylation

Lipid modifications, such as myristoylation, anchor eNOS to cellular membranes. eNOS is co-translationally myristoylated by N-myristoyltransferases, which irreversibly attach mirystic acid to the N-terminal glycine (Gly2) of eNOS (the N-terminal methionine is removed in this process) [55]. Subsequently, eNOS can be palmitoylated on cysteine residues (Cys15 and Cys26) on the Golgi apparatus, facilitating its trafficking to the PM caveolae [56]. Among other palmitoyltransferases, the Asp-His-His-Cys (DHHC)-motif-containing member 21 (DHHC-21) was identified as an important protein implicated in the palmitoylation of eNOS [57]. In contrast to myristoylation, posttranslational palmitoylation of eNOS is reversible, whereas eNOS depalmitoylation is catalyzed by acyl protein thioesterase-1 (APT-1) [58], an enzyme also involved in the hydrolysis of fatty acids linked to Ras proteins. In this regard, a similar cycle of palmitoylation–depalmitoylation has been proposed to regulate the intracellular distribution and activation of H- and N-Ras isoforms [59]. Remarkably, G2A mutants of eNOS lacking myristoylation and palmitoylation are randomly distributed in the cytosol and produce significantly lower amounts of NO [19,60]. Therefore, such modifications are important for the dynamic trafficking of eNOS, which may determine its enzymatic activity and intracellular function.

#### 2.2.2. Phosphorylation

Phosphorylation is one of the most studied PTMs of eNOS, rapidly modulating its enzymatic activity and interaction with other proteins [61]. Shear stress, H_2_O_2_, VEGF, insulin, and S1P are among the multiple stimuli that can induce the phosphorylation of eNOS, affecting several physiological and pathological processes [62,63,64,65,66]. Depending on the phosphorylated residues, the activity of eNOS can be upregulated or downregulated, impacting signal transduction. The phosphorylation of eNOS on Ser1177 by Akt, AMP-activated protein kinase (AMPK), protein kinase A (PKA), or CaM kinase II leads to its activation, enhancing NO production [67,68,69]. This site of phosphorylation has been extensively studied, as it promotes the activation of eNOS and its trafficking from caveolae to other compartments, such as the cytoskeleton [30].

More recent studies on the three-dimensional structure of eNOS have shown that phosphorylation on Ser1177 increases the activity of this enzyme by regulating its conformational dynamics [70], providing useful insight into the effect of phosphorylation on the activity of eNOS. Additionally, phosphorylation on Ser615 and Ser633 further stimulates eNOS activity in a Ca^2+^-independent way as a response to shear stress or bradykinin [71,72]. The phosphorylation of eNOS on Tyr83 by Src kinase in response to oxidative stress has also been identified as a target site of activation in bovine aortic endothelial cells [73]. On the other hand, phosphorylation on Thr495 by protein kinase C (PKC), Ser114 by extracellular signal-regulated kinase (ERK)-1/2, and Tyr657 by proline-rich tyrosine kinase 2 (PYK2) has been shown to inhibit eNOS activity [74,75].

#### 2.2.3. S-nitrosylation and Glutathionylation

S-nitrosylation has emerged as a paradigm of non-classical NO signaling [76]. eNOS can be S-nitrosylated on Cys94 and Cys98, reducing its enzymatic activity [77,78]. This reduction in activity is associated with the disruption of eNOS dimers, the active form of this enzyme; however, other cysteines susceptible to S-nitrosylation have been identified, suggesting that S-nitrosylation may regulate eNOS function in other ways aside from inducing monomerization [79,80]. Interestingly, S-nitrosylation appears to be closely linked to the subcellular localization of eNOS, as acylation-deficient eNOS mutants are defective in S-nitrosylation [78]. However, although several studies have reported the interesting finding that S-nitrosylation of eNOS may be linked to certain pathological states, such as endothelial or erectile dysfunction [81,82], further studies are required to underscore the relevance of protein S-nitrosylation in disease.

On the other hand, eNOS can also be glutathionylated (i.e., the covalent modification of Cys thiol groups with glutathione). Glutathionylation of eNOS in the C-terminal reductase domain (Cys689 and Cys908) has been shown to induce eNOS uncoupling [83], causing it to generate superoxide (O_2_^−^) and contribute to oxidative stress.

#### 2.2.4. Acetylation and Glycosylation

Either acetylation of eNOS on Lys607 or deacetylation on Lys497 and 4507 by the NAD^+^-dependent deacetylases SIRT1 and SIRT6 has been shown to increase eNOS activation by enabling CaM binding [84,85,86]. In contrast, other deacetylases have been identified, such as HDAC1 and HDAC3, which negatively regulate NO production by eNOS [87]. This regulation is particularly relevant in conditions where the function of the endothelium is compromised, such as metabolic diseases.

O-glycosylation is another nutrient-sensitive PTM modulating the activity of eNOS [54]. Pathological conditions, such as hyperglycemia and insulin resistance-induced excess fatty acid oxidation, increase endothelial concentrations of N-acetylglucosamine (GlcNAc) [88], leading to the attachment of GlcNAc to eNOS Ser1117. This modification has been shown to reduce Ser1177 phosphorylation and, thus, eNOS activity [89,90].

### 2.3. Protein–Protein Interactions

Ca^2+^-activated CaM was the first identified allosteric regulator of eNOS [91]. However, further research has uncovered a myriad of possible eNOS-interacting partners, which sometimes have opposing effects on its subcellular localization and enzymatic activity [30]. In this section, we outline some of the most important eNOS-binding proteins, sorted by their effects on the enzymatic function of eNOS.

#### 2.3.1. Positive Regulators of eNOS Activity

##### Calmodulin

Increasing intracellular Ca^2+^ levels cause CaM binding to eNOS, forming a tetramer of two CaM molecules bound to an eNOS dimer, facilitating electron transfer from the reductase to the oxidase domain, and inducing its dissociation from inhibitory proteins such as caveolin at the PM [16,92]. This Ca^2+^-dependent interaction is necessary for and proportional to the activity of eNOS. CaM itself can undergo phosphorylation in endothelial cells, attenuating eNOS activation [93]. Multiple pathways converge on the mobilization of intracellular Ca^2+^, causing the activation of eNOS via CaM, including agonists, such as bradykinin or acetylcholine [94]. Impaired eNOS dimerization and reduced NO bioavailability are characteristics of endothelial dysfunction, such as diabetes, where downregulation of CaM expression contributes to endothelial cell impairment [95,96]. However, several other stimuli, such as estrogens, ceramide, or shear stress, can increase eNOS activation without changing the level of intracellular Ca^2+^ [97,98,99]. Although these findings may seem contradictory, they highlight the importance of other Ca_2_^+^-independent PTMs for the functional regulation of eNOS.

##### Heat Shock Protein 90 (Hsp90)

Hsp90 is a molecular chaperone and activator of eNOS that stabilizes its structure and enhances its interaction with calmodulin [100]. In endothelial cells, the interaction between Hsp90 and eNOS is induced in response to stimuli such as shear stress, histamine, VEGF, and estrogens [101]. Hsp90 has a protective effect against calpain-mediated eNOS degradation, ensuring proper intracellular trafficking and activation [102]. Hsp90 has also been found to enhance eNOS function, facilitating the phosphorylation of eNOS by kinases such as Akt and balancing NO versus O_2_^−^ production [103,104]. Recent studies have described stromal cell-derived factor 2 (SDF2) as a critical factor for Hsp90-dependent eNOS activation, stimulating NO synthesis, and enhancing the phosphorylation of eNOS on Ser1177 [105]. In contrast, the C-terminus of Hsp70-interacting protein (CHIP), which is also an Hsp90 co-chaperone, has been shown to interact with eNOS, facilitating its translocation from the Golgi complex to a poorly characterized intracellular compartment where eNOS is inactivated [106].

Complexes between Hsp90, guanylate cyclase, and eNOS have been shown to promote the effective synthesis of cGMP [107]. However, this association has been found to be disrupted in pathophysiological situations. For instance, S-nitrosylation of the N-terminal region of Hsp90 disturbs its binding to eNOS, and reduced expression of Hsp90 in senescent endothelial cells is associated with eNOS uncoupling and is accompanied by defective NO production and formation of ROS and reactive nitrogen species (RNS) [108,109]. Additionally, hyperthermia-induced Hsp90–eNOS interactions have been shown to preserve mitochondrial respiration in hyperglycemia and alleviate the consequences of hypoxia in endothelial cells [110,111].

Altogether, these studies emphasize the key role played by the interaction between Hsp90 and eNOS in a broad range of endothelial dysfunction-related conditions.

##### G-Protein-Coupled Receptors (GPCRs)

Unlike Hsp90, some GPCRs interact with eNOS forming complexes that prevent its activation. The intracellular domain 4 (ID4) of the bradykinin 2 (B2) receptor was shown to negatively regulate eNOS in a manner similar to caveolin-1, as was demonstrated for the receptors of angiotensin II (Ang II) AT1 and endothelin B (ETB) [112]. It is known that agonist-stimulated phosphorylation of the ID4 region mediates GPCR and eNOS dissociation, thereby facilitating eNOS activation. Agonist binding of GPCRs induces the activation of heterotrimeric G-proteins. The GPCR pathways triggered by agonists like acetylcholine (m2 muscarinic receptor), bradykinin (B2 receptor), histamine, adenosine, ADP/ATP, sphingosine 1-phosphate (S1P), and thrombin are linked to Gαq or Gαi proteins, which activate phospholipase C and the release of intracellular Ca^2+^. Additionally, GPCRs are also linked to downstream kinases, such as PI3K/Akt, which trigger eNOS activation by phosphorylation on Ser1177 [94].

However, it has been shown that other GPCRs enhance the activity of eNOS through direct interaction, namely, GPCR kinase interactor-1 (GIT1) and β-Arrestin2, the latter being a critical component of the GIT1-eNOS signalosome that increases NO production by inducing eNOS phosphorylation [113,114]. β2-adrenergic receptors (β2AR) can also interact with eNOS and boost NO release via activation of protein kinase A (PKA) [115]. Moreover, numerous other studies have independently shown that G-protein-coupled estrogen receptors (GPER) improve cardiovascular function in postmenopausal women and arterial myography of healthy vessels through eNOS-dependent vasodilation [116]. Furthermore, under certain conditions, AT1 can increase eNOS activity in the presence of estrogens [117,118], suggesting that, for the regulation of eNOS, some interaction between AT1 and GPER pathways may exist.

##### Dynamin-2

Dynamin-2 is a large GTPase that directly binds to eNOS on the Golgi apparatus, although it can also co-localize with caveolin at the PM. Dynamin-2 enhances eNOS activity by increasing electron transfer and regulating eNOS trafficking between the PM and intracellular compartments [119,120,121]. Moreover, eNOS can also regulate and enhance the endocytosis of vesicles via the S-nitrosylation of dynamin [122]. Dynamin-2 is involved in different stages of the secretory pathway and has been shown to play a role in neuromuscular disorders as well as viral and bacterial infections [123,124,125,126]. In this regard, further evidence suggests that dynamin-2 and eNOS mediate the bacterial entry of uropathogenic *Escherichia coli* in bladder epithelial cells [127].

##### Cytoskeletal Proteins

eNOS also interacts with cytoskeletal proteins [128,129]. The binding of eNOS to G-actin facilitates eNOS phosphorylation, shifting its enzymatic production from O_2_- to NO [130,131]. The association of eNOS with the actin cytoskeleton can also occur indirectly through other eNOS interacting proteins, such as caveolin, dynamin, Hsp90, or NOS traffic inducer (NOSTRIN) (Figure 1), which interact with the actin nucleator-promoting factor neural Wiscott–Aldrich syndrome protein (N-WASP) [132,133,134,135]. Changes in the interaction between eNOS and actin have been linked to several physiological and pathological processes, such as endothelial cell growth or hypoxic situations [136,137,138]. Furthermore, the interaction of eNOS with G-actin is important for the activation of eNOS induced by agonists, such as thrombin, histamine, salbutamol, or adenosine [131]. In T cells, eNOS has been shown to be activated in response to antigen binding to the TCR. eNOS S-nitrosylates β-actin on Cys374, impairing β-actin binding to profilin-1 and actin polymerization at the immune synapse to increase the activation of PKC-θ [139].

Microtubules are also associated with eNOS. Pharmacological stabilization of microtubules has been shown to increase the association of eNOS with Hsp90, increasing basal eNOS phosphorylation on Ser1177 in a tubulin acetylation-dependent manner [140,141]. Furthermore, tubacin, a tubulin acetylation inducer that inhibits histone deacetylase 6 (HDAC6), was found to mitigate endothelial dysfunction by upregulating the expression of eNOS [142]. Regarding intermediate filaments, eNOS and vimentin have been confirmed to interact in the tight junctions of Sertoli cells [143]; although the features of this interaction remain to be investigated more in-depth, it has been suggested that NO may perturb the dynamics of tight junctions through a cGMP-dependent mechanism.

##### Porin (VDAC)

Voltage-dependent anion channels (VDACs) were originally characterized as mitochondrial porins, but later evidence indicated that they could also localize to the PM, where they interact with eNOS at the caveolae, participating in intracellular Ca^2+^ signaling and enhancing its activity [144]. Several human diseases have been associated with VDACs due to their roles in Ca^2+^ transportation and apoptosis. In the case of VDAC1 and VDAC2, their interaction with eNOS has been associated with protection against pulmonary hypertension [145,146].

##### Endoglin

Endoglin is a membrane glycoprotein abundantly produced in proliferating endothelial cells that plays an important role in angiogenesis [147]. Endoglin binds to eNOS at the PM caveolae, stabilizing its expression, facilitating its interaction with Hsp90, and promoting NO release [148]. Interestingly, endoglin haploinsufficiency has been associated with hereditary hemorrhagic telangiectasia type 1, which is linked to decreased bioavailability of eNOS-produced NO [149].

##### High-Density Lipoprotein (HDL) and Apolipoprotein AI (ApoAI)

High-density lipoprotein (HDL) cholesterol exerts direct effects on a variety of cell types, influencing cardiovascular and metabolic health, as well as anti-inflammatory and antioxidant actions in endothelial cells and leukocytes, among many other cell types [150,151,152]. ApoAI, a key HDL apolipoprotein, interacts with eNOS and promotes its phosphorylation, enhancing NO production [153]. In obese adolescents, restricted diet and exercise resulted in enhanced reverse cholesterol transport and the HDL-mediated activation of eNOS, positively influencing endothelial function; an action that could take place by endothelial lipase since it has recently been shown that endothelial lipase increases the eNOS-activating capacity of HDL and the cellular levels of esterified cholesterol [154,155]. Furthermore, low HDL cholesterol and the eNOS Glu298Asp polymorphism have been associated with inducible myocardial ischemia in patients with suspected stable coronary artery disease [156], supporting a role for these partners in providing enough blood flow to the heart.

##### Arginine Regulatory Enzymes

Reduced availability of L-arginine has been hypothesized as a primary cause of low eNOS activity and NO generation in cardiovascular disease, a hallmark of endothelial dysfunction. Notwithstanding the above, an “arginine paradox” has been described: even though intracellular levels of L-arginine are often sufficient for NO production by eNOS, the supplementation of L-arginine still enhances NO production. The arginine paradox may be due to the interaction of eNOS with arginine transporters and L-citrulline recycling enzymes, such as cationic amino acid transporter 1 (CAT1) and argininosuccinate synthase (ASS) or argininosuccinate lyase (ASL), respectively [157,158]. The capacity of CAT1 to control eNOS activation is thought not to be dependent on L-arginine transport but rather on allosteric regulation [159]. Interestingly, pertussis toxin-induced CAT1 activation in pulmonary artery endothelial cells (PAEC) boosted NO generation without affecting eNOS activity [160], implying that the caveola-associated CAT1–eNOS complex interacts with the L-arginine pool in endothelial cells to supply L-arginine to eNOS. The subcellular compartmentalization of arginine metabolizing enzymes has also been extensively studied [161], revealing their proximity to eNOS and the importance of access to L-arginine as a regulatory mechanism for the control of NO synthesis, especially in the context of endothelial dysfunction.

#### 2.3.2. Negative Regulators of eNOS Activity

##### Caveolin

Caveolin is a membrane protein closely linked to caveolae, which are 50–100 nm membrane microdomains in the PM composed of cholesterol, glycosphingolipids, and other structural and signaling proteins involved in cellular signaling and endocytosis [162,163]. Caveolin-1 binds to eNOS at the PM caveolae and inhibits its activity, blocking CaM binding [164]. Interestingly, recent studies have found that caveolin-1 can also be found in functional non-caveolar scaffold structures at the PM [165]. When intracellular Ca^2+^ levels rise in response to certain stimuli, caveolin-1 is displaced by Ca^2+^-CaM, increasing the enzymatic activity of eNOS [81]. Prolonged exposure to NO causes caveolin-1 nitrosation, phosphorylation, ubiquitination, and degradation [166,167], which may cause eNOS hyperactivation and uncoupling, increasing the levels of ROS from eNOS and also from NADPH oxidases since the latter are also inhibited by caveolin-1 [168].

In addition to caveolin-1, caveolin-2 and caveolin-3 can be found in caveolae isolated from endothelial cells and cardiac myocytes [169,170]. However, whereas caveolin-2 is unable to bind to eNOS, caveolin-3 is thought to interact with eNOS in cardiac myocytes.

In knockout mice, the loss of caveolin-1 gives rise to abnormal circulatory and pulmonary functions, and reduced expression of caveolin-1 is also associated with malignancy, fibrosis, and pulmonary hypertension [171,172,173]. Furthermore, genetic mutations or altered expression of caveolin-3 are known to cause muscular disorders, such as limb-girdle and Duchenne muscular dystrophy, a syndrome in which eNOS is poorly expressed [174]. This suggests that the interaction between caveolin-3 and eNOS would be important for muscular development. Atherosclerosis has also been reported to be exacerbated by either loss of eNOS-derived NO or overexpression of caveolin-1, being ameliorated by procedures boosting endothelial NO production, such as caveolin-1 deletion [175,176,177,178].

##### NOSIP and NOSTRIN

NOS-interacting protein (NOSIP) and NOSTRIN are eNOS-associated proteins that facilitate the translocation of eNOS from the PM to intracellular compartments, including the cytoskeleton and the Golgi apparatus (Figure 1) [179]. Interestingly, the relocation of eNOS to the cytoskeleton appears to be dependent on the cell cycle and the nucleocytoplasmic shuttling of NOSIP [180], which accumulates in the cytoplasm during the G2 phase.

In turn, NOSTRIN facilitates eNOS internalization from the PM to intracellular vesicles, reducing its enzymatic activity. The translocation of eNOS in this process involves a ternary complex of NOSTRIN with caveolin-1 [181]. NOSTRIN is a key adapter of a multimeric protein complex that also binds and regulates dynamin-2 and N-WASP, which are required for caveolar endocytosis and the internalization of eNOS by the actin cytoskeleton [182]. NOSTRIN may be involved in preeclampsia and portal hypertension associated with alcoholic hepatitis or cirrhosis [183,184,185], conditions in which high levels of NOSTRIN mRNA and protein have been associated with decreased eNOS activity.

##### Pin1

Pin1 is a prolyl isomerase that regulates the function of protein substrates through the isomerization of peptide bonds that link phosphoserine or phosphothreonine to proline, modulating multiple signaling pathways, some of which have been associated with the development of age-related vascular diseases [186]. Accordingly, Pin1 deficiency has been shown to cause endothelial dysfunction and hypertension, and its targeting can prevent mitochondrial oxidative stress and vascular dysfunction in patients with diabetes [187,188]. Regarding this, Pin1 has been shown to be associated with eNOS phosphorylated on Ser116, acting as a negative regulator of eNOS activity [189,190].

##### Plasminogen Activator Inhibitor-1

The role of fibrinolysis in the development of prediabetes-associated coronary heart disease was recently studied, indicating that there is a positive correlation between insulin resistance and plasminogen activator inhibitor-1 (PAI-1) during prediabetes [191]. Recently, PAI-1 was identified as an eNOS-interacting protein that co-localizes with eNOS on the Golgi and the perinuclear region of cultured human umbilical vein endothelial cells (HUVEC). PAI-1 was found to directly inhibit eNOS activity and reduce NO synthesis [192], placing PAI-1 as a novel negative regulator of eNOS in cardiovascular disease.

## 3. The Dynamic Intracellular Redistribution of eNOS

A large number of microscopy and biochemical studies, most of which have been based on the tracking of tagged eNOS in single cells and tissues, have shown that in response to stimuli such as shear stress, bradykinin, estrogen, or VEGF [193,194,195,196], eNOS dynamically redistributes among organelles within the cell. This trafficking is essential for regulating the activity of eNOS and the amount of NO produced [17,197]. In this section, we outline the cellular compartments where eNOS has been identified, as well as the function of this enzyme in each organelle.

### 3.1. eNOS in the Plasma Membrane and Caveolae

In endothelial cells, eNOS is mainly detected at the PM as well as the Golgi complex (Figure 1). At the PM, caveolae-associated eNOS is near ligand-accessible GPCRs, scavenging receptors, ion channels, kinases, and arginine transporters and recycling enzymes [197], allowing for rapid responses to extracellular signals and facilitating the release of NO into the extracellular space. Several studies have highlighted the role of palmitoylation in the targeting of eNOS towards the PM [198,199]. Robinson et al. identified an unusual pentameric dipeptide repeat (Gly-Leu)_5_, separating Cys15 and Cys26 eNOS palmitoylation sites. This sequence diverges from the well-characterized Met-Gly-Cys motif found in other dually acylated proteins, such as G-protein subunits and Src-family kinases [200].

The subcellular localization of eNOS within caveolae is dynamically regulated by agonists such as bradykinin, promoting enzyme depalmitoylation and eNOS translocation to other cell compartments [199]. eNOS activation by shear stress in resistance vessels also depends on its dissociation from caveolin-1 and translocation of the membrane-bound enzyme to the Golgi and cytosol [201]. Notably, the disruption of caveolin–eNOS complexes is mediated by agonist-induced Ca^2+^-CaM and Src kinase activation but not by the state of eNOS acylation [202]. In addition, in microvascular endothelial cells and mouse lung arteries, caveolae-mediated endocytosis has been shown to increase eNOS-derived NO in a manner dependent on dynamin-2, Gβγ, PI3K, and Akt signaling [203,204,205]. NO produced by eNOS raises Src activity and phosphorylates caveolin-1 Tyr14, which, in turn, enhances caveolar trafficking [206].

The regulation of eNOS by Ca^2+^ has been well established. However, recent reports have shed new light on the mechanisms by which Ca^2+^ fluxes may regulate eNOS activation at the PM. Activation of the Ca^2+^ channel TRPC5 by G-protein-coupled ATP receptors has been shown to induce Ca^2+^ influx and trigger NO production from eNOS, which sustains TRPC5 activation via cysteine S-nitrosylation [207]. Interestingly, the interaction between caveolin-1 and eNOS is disrupted by the activation of the ATP receptor, which may permit the association of eNOS with TRPC5, promoting Ca^2+^-mediated NO production. Another example of how Ca^2+^ fluxes could regulate eNOS at the PM involves capacitative Ca^2+^ entry (CCE) channels. CCE channels co-localize with eNOS in microdomains such as the caveolae of endothelial cells [208]. Recent computational modeling approaches have revealed that a subcellular interaction between eNOS and CCE channels significantly increases NO production as a result of higher Ca^2+^ intracellular fluxes.

More recently, it has been reported that the lipid composition of myoendothelial junctions (MEJ) (the PM interface between endothelial and smooth muscle cells in resistance arteries) allows the formation of two functionally distinct pools of eNOS in the endothelium, the MEJ itself and the blood-exposed apical domain [209]. eNOS in the apical endothelium is less dependent on Ca^2+^ fluxes than in the MEJ, which represents a new mechanism of compartmentalized eNOS activation in the blood vessel wall.

### 3.2. eNOS on the Golgi Apparatus

Early studies considered eNOS a Golgi-associated protein (Figure 1) [27]. The localization of eNOS in this organelle is thought to be important because the Golgi complex is a site for various PTMs, such as palmitoylation [57]. Phosphorylated eNOS on Ser1177 localizes in both the Golgi apparatus and the PM, indicating the existence of two cellular pools of active enzyme [196]. However, further studies revealed that eNOS constructs targeted towards the PM, although responsive to Ca^2+^, remained insensitive to activation by Akt-mediated phosphorylation; in contrast, eNOS constructs targeted to the *cis*-Golgi complex were less sensitive to Ca^2+^-dependent activation but strongly responsive to Akt-dependent phosphorylation [210]. Interestingly, targeting eNOS to the *trans*- and post-Golgi vesicles resulted in the production of very low amounts of basal NO by eNOS, demonstrating that compartmentalization is critical for the regulation of eNOS activation.

Hypertension, one of the risk factors of cardiovascular diseases, is sometimes associated with eNOS uncoupling and ROS production. Recent studies have measured O_2_^−^ and H_2_O_2_ in the Golgi of hypertensive mice, showing an association between an increased production of ROS in the Golgi and hypertension [211]. The specificity of eNOS targeting is also critical for PTMs and fine-tuned intracellular signaling. In this regard, it has been demonstrated that proteins are S-nitrosylated preferentially at the primary sites of eNOS localization, such as the Golgi, where S-nitrosylation of EMMPRIN and Golgi phosphoprotein 3 (GOLPH3) potentially modulates intracellular transport processes [212,213]. Beyond the functions of eNOS in the cardiovascular system, eNOS has been shown to regulate T cell receptor (TCR) signaling and activation from the Golgi complex of antigen-stimulated lymphocytes [11,214]. This process also involves the dynamic redistribution of eNOS, as the Golgi apparatus allocates to the immune synapse in these cells, permitting localized NO signaling [139]. Other cell types showing a compartmental distribution of eNOS are human embryonic stem cell-derived endothelial cells (hESC-ECs). hESC-ECs cultured in a fully defined medium revealed that lysophosphatidic acid (LPA) and platelet-activating factor (PAF) are also regulators of the localization of eNOS [215]. In these cells, eNOS is concentrated in the Golgi apparatus. However, treatment of hESC-ECs with LPA or PAF resulted in the redistribution of eNOS to the PM for its incorporation into caveolae-like structures. How LPA and PAF redistributed eNOS in hESC-ECs and whether this redistribution is dependent on palmitoylation in the Golgi apparatus is currently unknown.

### 3.3. eNOS in the Cytosol

Mutation of the N-myristoylation site of eNOS (Gly2) was shown to convert this membrane-attached enzyme into a cytosolic protein, and similar results were obtained under pharmacological conditions when cells were stimulated with high concentrations of bradykinin [19,216]. Although the implication of this localization has not been completely characterized, several studies have provided some clues to understand the possible functions exerted by eNOS in this subcellular compartment. Aberrant sequestration of eNOS in the cytoplasm has been described in endothelial cells treated with monocrotaline pyrrole (MCTP), a toxic plant alkaloid [217]. Immunofluorescence imaging revealed that MCTP induced the enlargement of PAEC, which was accompanied by a loss of eNOS from the PM and its accumulation in the cytoplasm, a feature also observed in hypoxia and senescence.

Recent reports have shown that cytosolic eNOS-derived NO plays a key role in the regulation of microvascular permeability [218,219]. The internalization of eNOS from the PM to the cytosol is a signaling mechanism that regulates hyperpermeability in platelet-activating factor (PAF)-induced inflammation [203]. Additionally, eNOS activated by PAF showed increased phosphorylation of Ser1177 and dephosphorylation of Thr495, increasing NO production and vascular permeability. Importantly, PAF failed to increase vascular permeability when eNOS was targeted at the PM, demonstrating that the cytosolic localization of eNOS is critical to this process [220]. Further studies on inflammation-induced hyperpermeability have revealed that both PAF and VEGF induce hyperpermeability in a self-limiting manner, stimulating the cAMP/Epac1 pathway to inactivate agonist-induced microvascular hyperpermeability. The process of hyperpermeability inactivation involves the vasodilator-stimulated phosphoprotein (VASP)-assisted translocation of eNOS from the cytosol to the PM of endothelial cells, becoming critical in the regulation of the inflammatory cascade [221]. Alternatively, S-nitrosylation has emerged as an important pathway for endothelial hyperpermeability [222,223]. The localization of eNOS in the cytosol was found to be fundamental to S-nitrosylate AJ proteins, such as VE-cadherin, ß-catenin, or p120 catenin (Figure 1), which are primary targets for PTMs, leading to PAF-induced endothelial hyperpermeability [224].

### 3.4. eNOS and the Endoplasmic Reticulum

The endoplasmic reticulum (ER) is the largest organelle in the cell and is a critical site for multiple cellular functions, including protein synthesis and transport, protein folding, lipid and steroid syntheses, glucose metabolism, and Ca^2+^ storage [225,226,227,228,229]. Regarding the localization of eNOS in this organelle, there are not many remarkable reports on the presence of eNOS in the ER. However, the activity of eNOS is critically modulated by intracellular Ca^2+^ fluxes from the ER (Figure 1). ER-released Ca^2+^ in endothelial cells increases phosphorylation of eNOS on Ser635 and NO production via ERK1/2 [230], a mechanism accounting for the regulation of eNOS activity by ATP and bradykinin.

Oxidized LDL (OxLDL) and oxysterols have been shown to play a crucial role in endothelial disfunction, inducing ER stress [231]. OxLDL and oxysterols promote eNOS uncoupling, causing the release of RNS and leading to nitrosative stress and aberrant protein S-nitrosylation, indicating their important role in cardiovascular diseases [232,233,234]. OxLDL can downregulate eNOS activity by causing ER stress, which may be implicated in the early stages of the pathogenesis of atherosclerosis [235]. Interestingly, oxLDL can induce S-nitrosylation of eNOS, enhancing the interaction between eNOS and β-catenin and influencing endothelial dysfunction [81].

### 3.5. eNOS on the Mitochondria

Since NO production was reported in the mitochondria, many research groups have tried to demonstrate the existence of a mitochondrial resident NOS isoform [236,237,238]. However, different known isoforms of NOS have been rather identified in the mitochondria of various cell types, suggesting that NOS proteins may translocate to this organelle to exert their physiological functions [239,240,241]. In this regard, recent findings have revealed that eNOS can interact with the mitochondria [242]. eNOS possesses a pentabasic amino acid sequence in its autoinhibitory domain, permitting its docking to the mitochondrial outer membrane and further supporting the possibility of this intracellular translocation (Figure 2). The mitochondrial redistribution of eNOS influences multiple cellular processes. For instance, inflammasome activation, induced by LPS, requires the mitochondrial redistribution of uncoupled eNOS, impairing mitochondrial bioenergetics and increasing the generation of mitochondrial ROS. Moreover, exercise training has been shown to increase eNOS expression and Ser1177 phosphorylation in the mitochondria of cardiomyocytes, identifying several of the key S-nitrosylated proteins involved in mitochondrial function and cardioprotection [243,244]. Furthermore, asymmetric dimethylarginine (ADMA) induces the translocation of eNOS to the mitochondria in ovine PAEC via protein nitration-dependent mechanisms [245].

The heat shock proteins Hsp90, Hsp70, and their co-chaperone CHIP—the eNOS interacting partners required for the folding and transport of precursor proteins towards the mitochondria—may also be involved in the redistribution of eNOS towards this subcellular compartment [246]. Further supporting this hypothesis, recent studies have shown that the enhanced activities of Hsp70 and CHIP in an animal model of pulmonary hypertension correlated with the increased presence of eNOS in the mitochondria [247,248].

The role of NO in the inhibition of mitochondrial respiration has long been studied [249,250,251]. Although there are circumstances, such as hyperthermia, where eNOS can preserve mitochondrial respiration, it is well accepted that NO competes with O_2_ for electrons from cytochrome c oxidase at complex IV of the electron respiratory chain of the inner mitochondrial membrane, producing signaling ROS [110,252]. In addition, the activity of eNOS-produced NO has been shown to be significant in the regulation of mitochondrial bioenergetics and biogenesis (reviewed in [253]), as observed in caloric restriction, where the extension of life span that induces the expression of eNOS was accompanied by increased oxygen consumption and ATP production [254]. On the other hand, mitochondrial eNOS may also have important implications in metabolism. The stimulation of endothelial cells with 17β-estradiol induced eNOS-dependent S-nitrosylation of the mitochondrial enzymes physiologically relevant for glucose and fatty acid oxidation [255,256], making evident new mechanisms by which mitochondrial-localized eNOS could regulate cell bioenergetics. Interestingly, p32, a protein implicated in mitochondrial function and cell metabolism, has been shown to regulate eNOS activation in endothelial cells by modulating the concentration of Ca^2+^ between the cytosol and mitochondria [257]. Moreover, the interaction of eNOS with the mitochondria was found to decrease oxidative stress in fetal PAEC, causing a reduction in O_2_ consumption and mitochondrial ROS production [258].

Although little is known about the pathophysiological actions exerted by mitochondrial eNOS in disease, it is worth noting that eNOS redistributes to the mitochondria in pulmonary endothelial disfunction and that both dimer formation and Akt-mediated phosphorylation are required for this process [259]. In contrast, simvastatin has been shown to restore pulmonary endothelial function by decreasing the mitochondrial redistribution of eNOS [259], highlighting the important regulatory function of mitochondrial eNOS for vascular lung permeability and inflammation.

### 3.6. eNOS in the Nucleus

Although the nuclear localization of eNOS has not been well explored, it has been reported that both estrogen- and VEGF-induced signaling cause the translocation of eNOS to the nucleus (Figure 2) [195,260]. In this regard, during angiogenesis, 17β-estradiol (E2) promotes the interaction of eNOS with the estrogen receptor (ER)α on the promoter of human telomerase reverse transcriptase (hTERT), inducing the transcription of the human telomerase gene [261]. On the other hand, VEGF promotes the nuclear translocation of eNOS and caveolin-1 with Flk-1/KDR and the VEGF receptor (VEGFR)-2 in endothelial cells [262]. Moreover, recent studies have revealed that the endosomal pathway is linked to the movement of VEGFR-2 from the Golgi apparatus to the nucleus [263]. Although the trafficking of eNOS was not assessed in this study, taking into account its localization on the Golgi apparatus and its involvement in VEGF-related pathways, similar mechanisms may be involved in the nuclear localization of eNOS.

More recently, an anti-apoptotic role for eNOS-produced NO in the nucleus has been reported. Aphidicolin, an inhibitor of DNA polymerases that represses DNA replication and leads to cell cycle arrest and damage, increases eNOS expression, phosphorylation on Ser1177, and nuclear localization, sustaining endothelial cell survival [264]. In addition, upon stimulation with VEGF, nuclear eNOS has been found to interact with and S-nitrosylate the RNA-binding protein ADAR1 (double-stranded RNA [dsRNA]-specific adenosine deaminase), influencing dsRNA stability and immune signaling pathways. When NO bioavailability is reduced, dsRNA accumulates, and type I interferon (IFN) responses are activated in pathogenic processes such as endothelial dysfunction and atherogenesis, which may promote plaque destabilization and the recruitment of monocytes to vascular lesions, ultimately leading to atherosclerosis progression [202,265,266].

**Figure 2 ijms-25-13402-f002:**
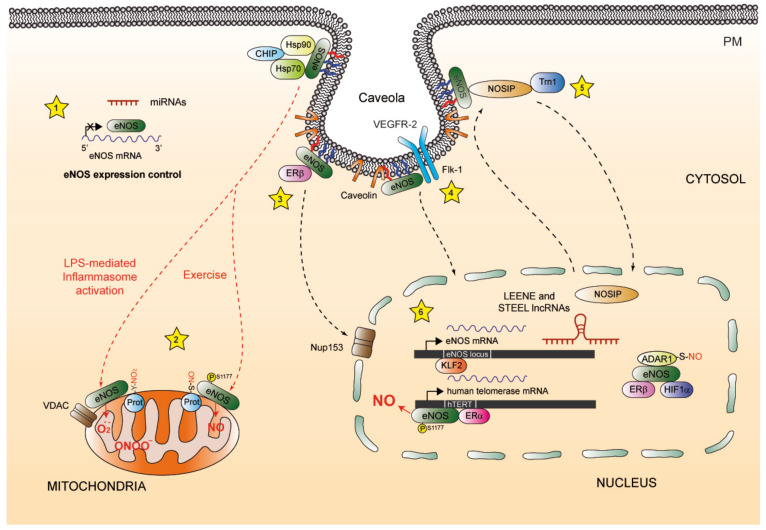
Trafficking and regulation of eNOS in the mitochondria and the nucleus. (1) The expression of eNOS is negatively regulated in the cytosol by eNOS-specific miRNAs. (2) eNOS can be translocated to the outer membrane of the mitochondria in a process that may involve molecular chaperones Hsp90, Hsp70, and CHIP. At this compartment, eNOS can synthesize both O_2_^−^ and NO, giving rise to peroxynitrite (ONOO^−^) and nitration and S-nitrosylation of mitochondrial proteins involved in metabolism and bioenergetics. (3 and 4) eNOS can be translocated to the nucleus by binding to the transcription factors ERβ or the complex VEGFR-2-Flk-1/KDR in response to estrogens and VEGF, respectively. (5) NOS-interacting protein (NOSIP) dynamically shuttles between the cytosol and the nuclear compartment, regulating eNOS activity. The nuclear translocation of NOSIP is mediated by transportin 1, a major nuclear import receptor [267]. (6) In the nucleus, eNOS binds to ERβ and hypoxia-induced factor-1 (HIF-1), promoting the transcriptional activity of human telomerase. In this compartment, eNOS also binds and S-nitrosylates ADAR1, which stabilizes double-strand RNA. On the other hand, long non-coding RNAs (lncRNA) increase the expression of eNOS in endothelial cells through the activation of the transcription factor Krüppel-like Factor 2 (KLF2).

Although many of the studies summarized above are not exempt from technical limitations, some of the latter sometimesrelated to the possible spatial restrictions conferred by the use of epitopes and large fluorescent proteins linked to eNOS and the modification of cell membranes and lipids by chemical fixation; they could be considered fundamental to our current understanding of the localization and function of eNOS in cells. Nevertheless, more analytical, real-time fluorescence microscopy studies in live cells (e.g., using super-resolution microscopy and single molecule tracking) would be required to address the localization and function of eNOS in-depth. Furthermore, advanced imaging studies such as those based on light-sheet fluorescence microscopy (LSFM) [268], a technology that permits 3D analysis of intact organs and organisms with low background and high contrast and sensitivity, would be necessary to further characterize in detail the trafficking and physiological function of eNOS.

## 4. Functional Implications of eNOS Subcellular Localization

As shown by the countless studies on the complex regulation of eNOS, its precise intracellular localization has important functional implications in various pathological conditions (Table 1).

Cardiovascular diseases are frequently linked to eNOS uncoupling and oxidative stress. In fact, the reduced bioavailability of NO has been considered a hallmark of endothelial dysfunction, which contributes to the pathophysiology of cardiovascular diseases, causing the development of risk factors such as hypertension or complications associated with diabetes [12,95]. Importantly, the mislocalization of eNOS from caveolae to other cellular compartments has been shown to increase its susceptibility to uncoupling [269]. Earlier studies showed that eNOS targeted to the mitochondria, nucleus, or *trans*-Golgi displayed reduced enzymatic activity compared to PM- and *cis*-Golgi-located constructs, indirectly affecting vascular tone in endothelial cells [270,271]. The subcellular localization of eNOS is thus essential for the maintenance of normal blood pressure, its dysregulation being associated with hypertension. Ang II has been shown to induce hypertension and endothelial dysfunction by multiple mechanisms involving eNOS signaling pathways [272,273]. Ang II increases superoxide production via the AT1 receptor through the activation of NADH/NADPH oxidase, disrupting eNOS-anchoring at the PM by oxidative stress [274]. Ang II signaling also reduces its association with caveolin-1 and modulates eNOS trafficking through downstream signaling pathways, slowing down the progression of disease [275,276].

The role of eNOS in atherosclerosis has been extensively studied [277]. Under physiological conditions, NO production by eNOS maintains vascular homeostasis by promoting vasodilation. However, factors like hypercholesterolemia and hypertension diminish NO levels, leading to impaired vasodilation and increased vascular stiffness [278,279]. Ang II signaling induces pro-inflammatory cytokine production and vascular smooth muscle cell proliferation, contributing to plaque development [280]. Additionally, several studies have highlighted the elevation of LDL cholesterol levels as an underlying factor of atherosclerotic cardiovascular disease [281]. In this regard, it is worth noting that OxLDL particles produced under oxidative stress have been shown to downregulate eNOS activity by displacing the enzyme from plasmalemmal caveolae, further promoting atherogenesis [235,282]. Diabetes is also a cause of endothelial and eNOS dysfunction [283]. Hyperglycemia has been shown to inhibit eNOS activity by increased O-linked GlcNAc modification, impeding its phosphorylation at Ser1177 and impairing NO production [89]. In the diabetic kidney, caveolin-1 localization at the plasma membrane was found to be decreased, altering eNOS trafficking and NO production inside the cell [284]. Importantly, insulin signaling modulates NO production, whereas elevated ROS in diabetes exacerbates eNOS dysfunction, causing further vascular complications [285,286].

Aside from the cardiovascular system, eNOS has also been associated with the development of neurological diseases, including neurodegeneration. In the nervous system, eNOS is considered to play dual roles, as its proper activation can enhance neuroprotection through improved blood flow and reduced oxidative stress, while its dysregulation may contribute to neuronal damage and disease progression [287]. In the case of brain disorders, the subcellular localization of eNOS may also represent a pathophysiological factor. The accumulation of eNOS at the hippocampal mitochondria of a scrapie-infected prion disease mouse model was found to partake in mitochondria dysfunction, which may influence the progression and histopathological changes observed in prion diseases [288]. Notably, eNOS dysregulation in neurons may disrupt mitochondrial respiration by O_2_^−^ release, influencing cell bioenergetics and leading to neurodegeneration [289]. Dysfunctional eNOS has also been shown to contribute to impaired neurovascular coupling, reduced amyloid clearance, and increased oxidative stress, which are hallmarks of Alzheimer’s disease and other forms of cognitive impairment [290].

The role of NO in cancer has recently been extensively reviewed elsewhere [291,292]. However, it is worth mentioning that eNOS-produced NO on the Golgi can contribute to tumor angiogenesis via the PI3K/Akt pathway, promoting N-Ras and H-Ras activation [293]. Furthermore, its activity has been linked to chronic inflammation, which can promote cancer-related processes, such as cell death resistance and metastasis [294]. On the other hand, pharmacological or genetic targeting of eNOS showed reduced tumor growth and progression in pancreatic and other K-Ras-driven cancers, supposedly through the regulation of H- and N-Ras on the Golgi and reducing NO levels near the plasma membrane, the subcellular domain of K-Ras, offering potential avenues for therapeutic interventions [295]. In prostate cancer, the translocation of eNOS to the nucleus has been identified as a relevant indicator of adverse clinical outcomes, a process in which nucleoporin 153 (Nup153) could play an important role since, recently, it has been identified as a regulator of the nuclear translocation of eNOS and ERβ in response to estrogen signaling [296]. In this context, estrogens induce the formation of complexes of eNOS with ERβ or HIF-1α in cancer tissues, leading to chromatin remodeling and transcriptional induction of prognostic genes, which highlights the participation of nuclear eNOS in epigenetic regulation.

**Table 1 ijms-25-13402-t001:** Physiological functions of eNOS and associated pathologies according to its subcellular localization.

eNOS Subcellular Localization	Physiological Function	Associated Pathology	References
Plasma membrane	Promotes vasodilation and blood pressure regulation in endothelial cells.	Hypertension, atherosclerosis, and endothelial dysfunction.	[12,198,209]
Golgi apparatus	Promotes vasodilation and blood pressure regulation. Modulates the function of Golgi resident proteins via S-nitrosylation.	Disruption of NO production at this location contributes to vascular remodeling and increased pulmonary vascular resistance.	[27,213,214]
Cytosol	Regulates microvascular permeability and endothelial barrier function.	eNOS translocation to the cytosol causes hyperpermeability in inflammation.	[18,217,221]
Endoplasmic reticulum	eNOS folding for trafficking to functional locations.	Endoplasmic reticulum stress under oxidative conditions leads to eNOS uncoupling in atherosclerosis.	[13,231,297]
Mitochondria	Regulates mitochondrial bioenergetics and cell metabolism.	Mitochondrial damage in neurodegenerative illnesses, such as prion diseases.	[242,245,288]
Nucleus	Regulates gene expression and cellular responses to hypoxia, VEGF, or estrogens.	Cancer progression and angiogenesis defects.	[261,262,266,296]

## 5. Conclusions and Future Directions

Given the importance of the functional distribution of eNOS, understanding the complex mechanisms behind subcellular targeting may be an essential step for elucidating its role in health and disease. In this regard, future studies aimed at using imaging-based spatial proteomics, a high-throughput methodology that combines high-resolution microscopy, proteomic analysis, and computational modeling [298], may be key to underscore the spatial localization and proteomic interactions of eNOS in single cells from tissues and organs. This relevant information would be important to decipher the contribution of the subcellular localization of eNOS to pathophysiological processes as a new parameter to consider in improving computational models for the understanding of the mechanisms of regulation of eNOS, such as it has been previously predicted for the angiogenesis inhibitor thrombospondin-1 (TSP1), a new variable controlling VEGF-induced eNOS signaling [299]. The dynamic distribution of eNOS among different cellular compartments, including the PM, Golgi apparatus, cytosol, mitochondria, and the nucleus, supports the notion that to regulate cellular signaling in a specific manner, NO production is fine-tuned in subcellular microdomains. Investigating how the localization of eNOS is disturbed in disease may provide new insights into the pathological mechanisms and potential therapeutic targets associated with endothelial dysfunction and altered NO signaling. In that respect, the development of CRISPR/Cas9 genome editing technologies and tissue-specific viral and non-viral delivering vector systems, which are currently seen as promising therapeutic tools [300], may provide new genetic approaches to target eNOS mutants to any cell compartment. The regulation of the production of NO at the subcellular level would provide new therapeutic opportunities for the treatment of cardiovascular-related diseases beyond the use of NO donors, whose limitations as therapeutic agents for the treatment of cardiovascular diseases include, among others, unfavorable hemodynamic effects, cytotoxicity, and low specificity and bioavailability [301]. However, further research is needed to fully characterize the functional significance of the localization of eNOS in other tissues and organs. A better understanding of the mechanisms involved in the subcellular trafficking of eNOS could thus give rise to significant therapeutic benefits, positively impacting the treatment of inflammatory diseases.

## Figures and Tables

**Figure 1 ijms-25-13402-f001:**
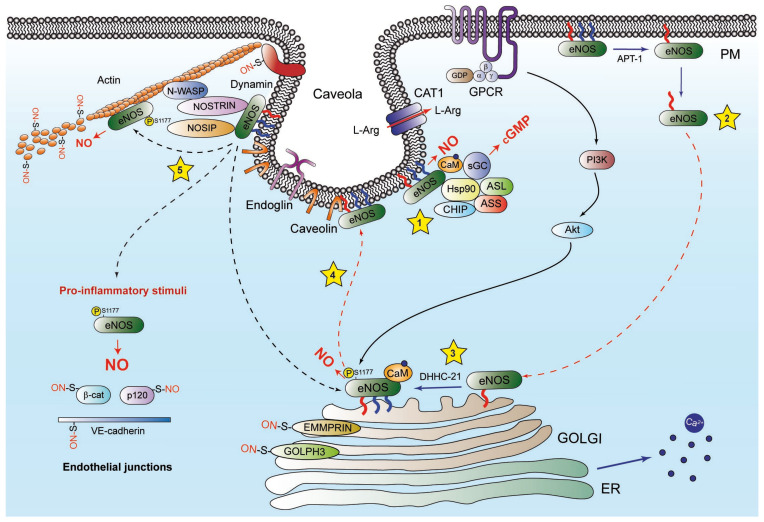
Trafficking and regulation of eNOS among the PM, the Golgi, and the cytosol. (1) eNOS associates with caveolin in plasma membrane (PM) caveolae, being activated upon receptor (e.g., GPCR)-induced Ca^2+^ release from the endoplasmic reticulum (ER) and subsequent binding to Ca^2+^-Calmodulin (CaM), which dissociates eNOS from caveolin. The activation of eNOS by Ca^2+^-CaM is stabilized by heat shock protein 90 (Hsp90) and stimulated by L-Arg transporter CAT1 and L-citrulline recyclers argininosuccinate lyase (ASL) and argininosuccinate synthase (ASS), being the soluble guanylate cyclase (sGC) and the production of cGMP as its classical signaling effectors. (2-4) eNOS trafficks from the PM to the Golgi apparatus by acyl protein thioesterase 1 (APT-1)-mediated depalmitoylation and returns to the PM caveolae by DHHC21-mediated repalmitoylation (palmitoylation displayed in blue and myristoylation in red). On the *cis*-Golgi, eNOS is mainly activated to produce NO by PI3K-Akt-mediated phosphorylation on Cys1177, S-nitrosylating Golgi resident protein transporters such as EMMPRIN and Golgi phosphoprotein 3 (GOLPH3). (5) eNOS binding to NOS-interacting protein (NOSIP) and NOS traffic inducer (NOSTRIN) at the PM is also important for eNOS trafficking to the Golgi complex and other subcellular compartments, such as the actin cytoskeleton and the cytosol, where the activation of eNOS can promote S-nitrosylation-mediated F-actin depolymerization and disruption of endothelial junctions; the former through S-nitrosylation of β-actin and the latter by S-nitrosylation of β-catenin, VE-cadherin, and p120 catenin.

## Data Availability

Not applicable.

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
