# Peer review of "Subcellular Localization Guides eNOS Function"

_ijms, 2024, doi:10.3390/ijms252413402_

Round 1

Reviewer 1 Report

Comments and Suggestions for Authors

MS number: 33248

Title: Subcellular Localization Guides eNOS Function

Authors: Leticia Villadangos and Juan M. Serrador

Summary:
The current review explores the regulation of endothelial nitric oxide synthase (eNOS), underlining its subcellular localization and its influence on nitric oxide (NO) production. NO is a crucial signaling molecule with multiple molecular targets, and its bioavailability is regulated through eNOS localization across subcellular compartments, including the plasma membrane, Golgi apparatus, mitochondria, and nucleus.

The article systematically details mechanisms driving eNOS localization, such as post-translational modifications (e.g., palmitoylation, phosphorylation), protein-protein interactions, and responses to physiological stimuli. Additionally, it connects eNOS mislocalization to pathological conditions, particularly cardiovascular diseases, and discusses the potential for therapeutic targeting. The review highlights how these processes control eNOS enzymatic activity and NO production, emphasizing the pathological consequences of dysregulation, particularly in cardiovascular diseases such as hypertension, atherosclerosis, and diabetes.

By detailing historical insights and cutting-edge research, the review underscores the importance of eNOS’s spatial distribution in determining its functional outcomes. It further calls for continued investigation into the mechanisms of eNOS trafficking, with an emphasis on the broader physiological and pathological contexts and potential therapeutic interventions targeting its compartmentalization.

The manuscript is comprehensive and rooted in current literature, however, it would benefit from expanding the discussion on the non-cardiovascular roles of eNOS and further implications in broader physiological and pathological contexts.

Strengths:

  1. Detailed overview of the molecular mechanisms regulating eNOS activity.
  2. Thorough integration of foundational studies with recent discoveries.
  3. Well-referenced discussion of NO signaling pathways and implications in cardiovascular diseases.
  4. There is a clear connection between eNOS localization, enzymatic activity, and NO signaling.

Weaknesses:

  1. Limited exploration of eNOS roles in tissues beyond the vascular system.
  2. There is minimal discussion on translational strategies to modulate eNOS trafficking for therapeutic purposes.
Comments on the Quality of English Language

The manuscript demonstrates a high level of linguistic proficiency, with clear and well-structured scientific language. The vocabulary is precise, and the technical terms are used appropriately.

However, a careful review of spelling, grammar, and style consistency is essential for final publication. Read the text carefully to account for missing articles or prepositions.

Here are specific observations:

Strengths:

The manuscript makes effective use of specialist terminology, such as "subcellular compartmentalization," "post-translational modifications," and "eNOS localization," enhancing its readability for a scientific audience. The logical organization of sections and subsections ensures readability and coherent argumentation. The abstract and conclusion succinctly summarize the content, emphasizing the review's objectives and significance.

Areas for Improvement:

Sentence Structure

Some sentences are overly long or complicated, potentially hindering readability. Breaking them into shorter, more concise statements can improve clarity.

325-327; 448-451; 544-547

Spelling Issues

While the manuscript is generally free of spelling errors, there are a few places where technical terms could benefit from precise formatting:

Example: Misused Terms or Inconsistent Use of Hyphens:

  • line 552 - "endoplasmic reticulum stress" is correct but sometimes abbreviated inconsistently (line 556 - e.g., "ER stress"). Always ensure the abbreviation is defined and consistently used.

The manuscript appears free of spelling mistakes; however, a few minor grammatical adjustments may enhance clarity. For example:

·        Line 37 “...highlighting its importance in important pathophysiological processes...” — Replace "importance in important" with more varied language to avoid redundancy.

Style and Consistency:

Authors should use consistent abbreviations once they are defined, e.g., "eNOS" and "NO," without unnecessarily reverting to the full forms.

Citation Placement:

Authors should place quotes appropriately and not disrupt the flow of the sentences.

Recommendations:

Authors should perform final proofreading to correct any remaining typographical errors and ensure consistency of style.

Authors should also use tools such as Grammarly or language editing services to detect subtle language errors or work with a professional language editor experienced in writing scientific manuscripts for further polish. 

Conclusion:

These examples demonstrate the need to improve sentence clarity, grammatical precision, and consistency throughout the manuscript. A systematic review will improve readability and professional presentation.

Author Response

Reviewer #1

We very much appreciate the constructive reviewer’s comments, which have all been considered to improve the manuscript.

Weaknesses: 

  1. Limited exploration of eNOS roles in tissues beyond the vascular system.

-We agree with the reviewer that the manuscript focuses on the role of eNOS subcellular localization in endothelial cells and the cardiovascular system. However, we have also added examples highlighting the localization and function of eNOS in T lymphocytes, hematopoietic stem cells and cardiomyocytes – page 7, lines 271-274, page 12, lines 460-464 and 464-472, and page 15, lines 548-551. It has also been reported that other cells, such as erythrocytes and platelets, also express eNOS. Nonetheless, the absence of intracellular organelle organization in these cells makes them less appropriate for the type of cell biology studies performed in endothelial cells. Moreover, endothelial cells are the prototypic eNOS-expressing cells and are thus suitable cellular models for the study of eNOS localization and function in pathophysiological processes. Therefore, findings from their studies can be translated to other cell systems expressing eNOS.

The role of eNOS in pathologies beyond the cardiovascular system has also been discussed. We have added a new section to the manuscript, entitled “Functional implications of eNOS subcellular localization” – pages 18-19 –, which delves into the relevance of eNOS subcellular localization in various diseases, including neurodegeneration and cancer - page 19, lines 677-707.

  1. There is minimal discussion on translational strategies to modulate eNOS trafficking for therapeutic purposes.

-The possibility of using the localization of eNOS as a therapeutic tool to specifically deliver NO to the subcellular compartments where it is needed may represent a sophisticated interventional approach for the rectification of NO subcellular levels, especially in diseases in which the localization of eNOS can play an important role, such as hypertension or atherosclerosis. However, we think that these approaches are far from being implemented nowadays. We have dealt with this issue in the manuscript section 5 “Conclusions and future directions” – pages 20-21, lines 739-747. Importantly, CRISPR/Cas9 genome editing technologies and tissue-specific viral and non-viral delivering vector systems should be thoroughly explored for therapeutic purposes to control eNOS function or subcellular localization, as it is implicated in various pathophysiological processes.

Comments on the Quality of English Language

Sentence Structure

Some sentences are overly long or complicated, potentially hindering readability. Breaking them into shorter, more concise statements can improve clarity.

325-327;

-We have rewritten this sentence for clarity.

448-451;

-We have rewritten this sentence for clarity.

544-547

-This sentence has been deleted since we have realized it is not essential in the context of the main paragraph.

Spelling Issues

While the manuscript is generally free of spelling errors, there are a few places where technical terms could benefit from precise formatting:

Example: Misused Terms or Inconsistent Use of Hyphens:

  • line 552 - "endoplasmic reticulum stress" is correct but sometimes abbreviated inconsistently (line 556 - e.g., "ER stress"). Always ensure the abbreviation is defined and consistently used.

- We have revised the abbreviations and avoided inconsistencies in the manuscript. In this case, the “endoplasmic reticulum” was abbreviated as “ER” after its first apparition in the text.

The manuscript appears free of spelling mistakes; however, a few minor grammatical adjustments may enhance clarity. For example:

  • Line 37 “...highlighting its importance in important pathophysiological processes...” — Replace "importance in important" with more varied language to avoid redundancy.

- Thank you for this correction. We used the word “key” instead of “important” in the second part of the sentence.

Style and Consistency:

Authors should use consistent abbreviations once they are defined, e.g., "eNOS" and "NO," without unnecessarily reverting to the full forms.

-As in the case of the “ER,” we have revised the use and definition of the most common abbreviations in the manuscript.

Citation Placement:

Authors should place quotes appropriately and not disrupt the flow of the sentences.

-We have thoroughly revised the position of references in the manuscript sentences.

Recommendations:

Authors should perform final proofreading to correct any remaining typographical errors and ensure consistency of style.

Authors should also use tools such as Grammarly or language editing services to detect subtle language errors or work with a professional language editor experienced in writing scientific manuscripts for further polish.

-Following the helpful recommendations of the reviewer, we have performed a deep proofreading of the manuscript and have worked with a professional language editor to minimize mistakes and improve the writing style.

Reviewer 2 Report

Comments and Suggestions for Authors

Key points:

Comprehensive review: Covers mechanisms such as post-translational modifications, protein-protein interactions, and signaling pathways for eNOS localization.

Multidisciplinary relevance: applicable to cardiovascular, metabolic and inflammatory diseases.

Visual representation: Figures, such as interorgan traffic patterns, clarify complex processes. Including more illustrated figures would improve the manuscript and make it easy for the readers.

Integration of emerging findings: emphasizes the role of RNA-binding proteins, noncoding RNAs, and new connections such as mitochondrial eNOS in bioenergetics and immune signaling.

Weakness: Overemphasis on established concepts: Repeats known mechanisms such as the regulation of calcium and calmodulin without providing new insights.

Limited discussion of disease states: Underexplores diseases such as neurodegeneration and cancer and lacks specificity in the relationship between conditions such as atherosclerosis and diabetes.

Methodological gap: it relies heavily on the existing literature without addressing the limitations of the mentioned studies, such as the real-time visualization of eNOS traffic.

Insufficient coverage of therapeutic potential: unclear on potential interventions targeting eNOS localization and lack of discussion on pharmacological modulators or gene modification.

Future directions:

Integrating technology: highlighting advances in live cell imaging and proteomics for eNOS dynamics. Therapeutic Translation: propose strategies to modulate eNOS localization for therapeutic benefit. Systems biology approach: using computational modeling to predict eNOS behavior under different conditions.

Comments on the Quality of English Language

Please check for any mistakes.

Author Response

Reviewer #2

We thank the reviewer for his/her effort in reviewing our manuscript and acknowledge the comments and suggestions provided. We have performed the changes indicated and agree that the manuscript is now improved for publication in IJMS.

Weakness: Overemphasis on established concepts: Repeats known mechanisms such as the regulation of calcium and calmodulin without providing new insights.

-Following the reviewer’s instructions, in the new version of the manuscript we have tried to minimize the number of times that common mechanisms of eNOS regulation are repeated along the manuscript sections. For this purpose, we have deleted a large number of sentences, which in one way or another dealt with the same issue - lines 40-41, 43-45, 51-52, 106-108, 119-121, 230-232, 236-238, 247-244, 398-408, 434-435, 440-443, 447, 544-545 and 609-610 of the former version of the manuscript, and other sentences have been rewritten for better clarity. However, the reviewer should realize that some of these regulatory mechanisms, in particular those involving the interaction between Ca2+ and calmodulin, are essential to understand the regulation of eNOS in almost every context and are hard to avoid when some studies are commented. Regarding the reviewer’s comment that new insights are not provided on this topic, we would like to note that the manuscript provides some new insights on the mechanisms by which Ca2+ can regulate the activity of eNOS, particularly those related to the involvement of transient receptor potential channel 5 (TRPC5), capacitative Ca2+ entry (CCE) channels and Ca2+ fluxes in the regulation of eNOS at the PM – page 11, lines 421-432. We think that these findings are new and worth to be highlighted in the manuscript.

-Limited discussion of disease states: Underexplores diseases such as neurodegeneration and cancer and lacks specificity in the relationship between conditions such as atherosclerosis and diabetes.

Regarding this interesting point, a new section 4: “Functional implications of eNOS subcellular localization” has been added to the manuscript – pages 18-19. In this new section, the relevance of the subcellular localization of eNOS in neurodegeneration and cancer has been discussed, as shown on page 19, lines 677-707. However, the latter has been more succinctly treated, as several excellent reviews on this topic have been recently published (e.g. Navasardyan and Bonavida, Cells 2021, Ref. 294; Tang et al., J. Nanobiotechnology 2024, Ref. 293). The relationship between eNOS function, atherosclerosis and diabetes has also been addressed, revealing the role of hypercholesterolemia or hyperglycemia in eNOS function and the pathophysiology of these diseases – pages 18-19, lines 660-676. In addition, a table (Table I) representing the localization, function and involvement of eNOS in disease and the corresponding publications supporting this function has been now included at the end of this section – page 20. 

Methodological gap: it relies heavily on the existing literature without addressing the limitations of the mentioned studies, such as the real-time visualization of eNOS traffic.

The lack of analytical live imaging studies of eNOS trafficking in cells, organs and organisms to study the actual relevance of the localization of eNOS in physiological and pathological processes, together with some technical limitations inherent to the microscopy and biochemical studies addressed in the manuscript, have been now discussed at the end of the section 3: “The dynamic intracellular redistribution of eNOS” – pages 17-18, lines 627-638.

Insufficient coverage of therapeutic potential: unclear on potential interventions targeting eNOS localization and lack of discussion on pharmacological modulators or gene modification.

The possibility of using the localization of eNOS as a therapeutic tool to specifically deliver NO to the subcellular compartments where it is needed may represent a sophisticated interventional approach for the rectification of NO subcellular levels in some disease conditions in which the localization of eNOS can play a role, such as hypertension and atherosclerosis. However, we think that these approaches are far from being implemented nowadays. As the reviewer has suggested in his next comment, we think this issue should be more appropriately treated as “future directions”. 

Future directions:

Integrating technology: highlighting advances in live cell imaging and proteomics for eNOS dynamics. Therapeutic Translation: propose strategies to modulate eNOS localization for therapeutic benefit. Systems biology approach: using computational modeling to predict eNOS behavior under different conditions.

The new available technical advances to study the localization of eNOS by a combination of live imaging and proteomics, the promising therapeutic strategies based on genetic approaches to modulate the activation of eNOS in subcellular compartments for therapeutic benefits in the treatment of cardiovascular diseases, and the use of computational modelling to understand the regulation and function of eNOS at specific cells compartments have all been discussed in the section 5: “Conclusions and future directions” – pages 20-21, lines 724-733 and 739-747.

 Visual representation: Figures, such as interorgan traffic patterns, clarify complex processes. Including more illustrated figures would improve the manuscript and make it easy for the readers.

In addition to the new - Table I, for further clarification of the processes taking part in eNOS trafficking, we have splitted the former Figure 1 into two new figures, which show the relationship between eNOS trafficking and post-translational modifications in the cytosol and cell membranes - Figure 1, Page 13, and on the mitochondria and the nucleus - Figure 2, Page 17.

 Comments on the Quality of English Language

Please check for any mistakes.

We have performed a careful proofreading of the manuscript in order to correct any possible English language mistakes.

Reviewer 3 Report

Comments and Suggestions for Authors

This review article discusses the role of endothelial NO synthase, how the subcellular localization controls the activity of the enzyme and bioavailability of NO. It reviews basic and recent studies which point towards mechanism or factors that influence the trafficking and functional dynamics of eNOS across membrane, Golgi apparatus, cytoplasm, mitochondria, and nucleus.

The importance of the article is underlined by the fact that it highlights how intracellular trafficking of eNOS influences NO output and signaling at the point of its implications for eNOS dysfunction in diseases, with particular emphasis on cardiovascular disorders. Moreover, it points out future therapeutic perspectives on eNOS subcellular localization.

In my opinion the article could be published if some major issues are addressed.

Introduction: There is good background, but it would be even more useful to summarize in one sentence the critical features of eNOS subcellular localization. Perhaps technical details, such as substrates and coenzymes required, could be placed in specialized sections.

eNOS Regulatory Mechanisms: The sections on post-translational modifications and protein interactions are quite comprehensive, but somewhat summarizing could have been done. Features such as palmitoylation and myristoylation may be summarized to just state the key effects of the modifications on enzyme activity.

Subcellular Compartmentalization: It is a very useful compartmentalization, but somewhat redundancy-ridden, into sub-compartments such as membrane, Golgi, and cytoplasm. The section on Golgi may be easily integrated into sections on modifications taking place in the Golgi to avoid duplication of material.

Simplification of Technical Terms: Some highly technical explanations, such as the minute level of enzymes involved in the acylation process or microRNAs, need to be made more understandable for a wide audience.

Figures: Include a scheme or figure explaining in detail eNOS trafficking across compartments together with its post-translational modifications.

Author Response

Reviewer #3

We thank Reviewer #3 for his/her constructive comments which have been taken into account to improve the study and to make the review more cohesive and accessible to a broad non-specialist audience. We answer his/her comments and suggestions below.

Introduction: There is good background, but it would be even more useful to summarize in one sentence the critical features of eNOS subcellular localization. Perhaps technical details, such as substrates and coenzymes required, could be placed in specialized sections.

As suggested by the reviewer, we have now summarized in one sentence the most important features of the subcellular characterization of eNOS in the Introduction section – page 2, lines 49-53. However, unless the reviewer considers otherwise, we would like to retain some important technical details in this initial section, such as the common characteristic reaction of nitric oxide synthases – page 1, lines 23-27-, as we consider it essential to define this family of proteins and their enzymatic activity. Furthermore, we think this information does not fit in other sections of the manuscript, as they are mainly focused on eNOS dynamics.

eNOS Regulatory Mechanisms: The sections on post-translational modifications and protein interactions are quite comprehensive, but somewhat summarizing could have been done. Features such as palmitoylation and myristoylation may be summarized to just state the key effects of the modifications on enzyme activity.

In this new version of the manuscript, we have tried to summarize the most important post-translational modifications of eNOS, particularly palmitoylation and myristoylation - pages 3-4, lines 113-129. However, the reviewer should realize that these kinds of modifications play a key role in the mechanisms involved in the subcellular distribution of eNOS and thus, are of great relevance for the technical development of genetic procedures leading to possible eNOS localization-based therapeutic approaches for the treatment of disease. For this reason, we would like to describe them in some detail.

Subcellular Compartmentalization: It is a very useful compartmentalization, but somewhat redundancy-ridden, into sub-compartments such as membrane, Golgi, and cytoplasm. The section on Golgi may be easily integrated into sections on modifications taking place in the Golgi to avoid duplication of material.

We agree with the reviewer that some common aspects of eNOS post-translational modifications and the regulation of compartmentalization (e.g. acylation and Golgi localization) were repeated in the former version of the manuscript. In this new version, we have tried to correct this issue, eliminating as many redundant paragraphs as possible – lines 40-41, 43-45, 51-52, 106-108, 119-121, 230-232, 236-238, 247-244, 398-408, 434-435, 440-443, 447, 544-545 and 609-610 of the former version of the manuscript, and rewriting some sentences for better clarity. However, the reviewer must be aware of the importance of the post-transcriptional modifications for the subcellular localization of eNOS and the difficulty of discussing one of these regulatory mechanisms without addressing the other. Upon these circumstances, it is unavoidable not to mention some important aspects of palmitoylation and myristoylation in the Golgi section. On the other hand, although the integration of the Golgi section into the sections on modifications taking place in the Golgi may be an appropriate way to avoid duplication of material, given the importance of subcellular compartments in the main topic of the review, we would like to maintain the corresponding Golgi section with some modifications.

Simplification of Technical Terms: Some highly technical explanations, such as the minute level of enzymes involved in the acylation process or microRNAs, need to be made more understandable for a wide audience.

In this new version of the manuscript, we have included additional clarifications, such as definitions and practical aspects or applications, in order to simplify technical explanations of processes, enzymes and non-coding RNAs involved in the cellular redistribution of eNOS – page 3, lines 85-93 and lines 118-120. We hope that this new information will be helpful for a general reader, and make this part of the manuscript more understandable.

Figures: Include a scheme or figure explaining in detail eNOS trafficking across compartments together with its post-translational modifications.

For further clarification of the processes taking part in eNOS trafficking, we have splitted the former Figure 1 into two new figures, which show the relationship between eNOS trafficking and post-translational modifications in the cytosol and cell membranes - Figure 1, page 13, and on the mitochondria and the nucleus -Figure 2, page 17. We have also added a new table at the end of section 4 – page 20 – highlighting the functions of the subcellular localization of eNOS in important disease-associated processes.

Round 2

Reviewer 3 Report

Comments and Suggestions for Authors

The authors made all corrections. The revised version is completely agreeable to me, for this reason, I recommend the publication of the manuscript.